# Anthelmintic Properties of Essential Oils to Control Gastrointestinal Nematodes in Sheep—In Vitro and In Vivo Studies

**DOI:** 10.3390/vetsci9020093

**Published:** 2022-02-19

**Authors:** Filip Štrbac, Antonio Bosco, Maria Paola Maurelli, Radomir Ratajac, Dragica Stojanović, Nataša Simin, Dejan Orčić, Ivan Pušić, Slobodan Krnjajić, Smaragda Sotiraki, Giorgio Saralli, Giuseppe Cringoli, Laura Rinaldi

**Affiliations:** 1Department of Veterinary Medicine, Faculty of Agriculture, University of Novi Sad, Trg Dositeja Obradovića 8, 21102 Novi Sad, Serbia; dragicas@polj.edu.rs; 2Department of Veterinary Medicine and Animal Production, University of Naples Federico II, CREMOPAR, Via Federico Delpino 1, 80137 Naples, Italy; boscoant@tiscali.it (A.B.); mariapaola.maurelli@unina.it (M.P.M.); giuseppe.cringoli@unina.it (G.C.); lrinaldi@unina.it (L.R.); 3Scientific Veterinary Institute Novi Sad, Rumenački put 20, 21113 Novi Sad, Serbia; ratajac@niv.ns.ac.rs (R.R.); ivan@niv.ns.ac.rs (I.P.); 4Department of Chemistry, Biochemistry and Environmental Protection, Faculty of Sciences, University of Novi Sad, Trg Dositeja Obradovića 3, 21102 Novi Sad, Serbia; natasa.simin@dh.uns.ac.rs (N.S.); dejan.orcic@dh.uns.ac.rs (D.O.); 5Institute for Multidisciplinary Research, University of Belgrade, Kneza Višeslava 1, 11030 Belgrade, Serbia; slobodan.krnjajic@imsi.bg.ac.rs; 6Veterinary Research Institute, National Agricultural Research Foundation, NAGREF Campus, 57001 Thessaloniki, Greece; sotiraki@vri.gr; 7Experimental Zooprophylactic Institute of Lazio and Tuscany M. Aleandri, Via Appia Nuova, 00178 Rome, Italy; giorgio.saralli@izslt.it

**Keywords:** phytotherapy, essential oils, anthelmintic efficacy, gastrointestinal nematodes, in vitro test, in vivo test

## Abstract

Herbal products such as essential oils may play a promising role in the treatment of infections caused by gastrointestinal nematodes (GINs). The aim of this study was to evaluate the in vitro potential of 11 essential oils (EOs) and one binary combination of isolated EO compounds, as well as the in vivo anthelmintic efficacy of two EO formulations. Four GIN genera were identified in the coproculture examination: *Haemonchus*, *Trichostrongylus*, *Teladorsagia* and *Chabertia*. The in vitro egg hatch test (EHT) was performed at six different concentrations (50, 12.5, 3.125, 0.781, 0.195 and 0.049 mg/mL) for each EO, whereas in the in vivo faecal egg count reduction test (FECRT), each EO sample was diluted in sunflower oil and orally administrated at a dose of 100 mg/kg to the different group of animals. In the EHT, the EOs of *Origanum vulgare*, *Foeniculum vulgare*, *Satureja montana*, *Satureja hortensis* and two types of *Thymus vulgaris* were the most effective. The dominant compounds of these EOs were carvacrol, thymol, anethol, p-cymene and γ-terpinene, indicating their importance for the anthelmintic activity. In the FECRT, both *T. vulgaris* EO type 1 and linalool:estragole combination show an anthelmintic potential with a mean effect on FECR of approximately 25%. The results suggest the possible role of tested EOs as anthelmintic agents in sheep farms, although further in vivo tests are needed.

## 1. Introduction 

Gastrointestinal nematodes (GINs) are still the most prevalent parasites that cause disease in grazing ruminants worldwide, particularly in sheep and goats [1]. The prevalence of certain GIN genera depends on geographic and climate conditions. Generally, the most prevalent genera are *Haemonchus*, *Trichostrongylus*, *Teladorsagia*, *Chabertia*, *Cooperia*, *Nematodirus* and *Oesophagostomum* [2,3,4,5,6]. In most cases, infections with these parasites manifest in the form of a subclinically impaired weight gain as well as an impaired yield of meat and milk, but can also seriously endanger animal health and welfare by causing conditions such as diarrhoea, digestive problems, anorexia, anaemia, protein loss as well as reduced immunity, which can lead to death [1,4,7]. For these reasons, GINs are responsible for huge economic and productive losses around the world. Although these losses are difficult to estimate, it is regarded that they amount to 17.94% of the total economic cost in animals [8], or hundreds of millions of euros per year across Europe [9].

The management of GIN infections represents a challenge that currently depends almost exclusively on commercial anthelmintic preparations [4]. These drugs were proven effective in the control of GINs in previous decades and make up to 53% of total used veterinary drugs [8,10]. However, excessive use, often resulting in overfrequent treatments, has led to the development of anthelmintic resistance (AR), which has been reported in many parts of the world for different nematode species [4,11]. Consequently, the declined efficacy of most commonly used anthelmintic drug classes such as benzimidazoles (BZ) and macrocyclic lactones (ML) has caused additional high economic losses, which overall endanger modern livestock [4,12]. This justifies the urgent need for new, alternative strategies such as genetic control methods, pasture management, nutritional manipulation, biological regulation (such as the use of nematophagus fungi), vaccine development and the use of different herbal-based products [11,13].

In recent years, botanical anthelmintics have emerged as an promising alternative in the control of ruminant GINs and as a tool to combat AR [8,14,15,16,17]. Among other plant products, essential oils (EOs) are the most commonly mentioned and examined within that context [18]. EOs represent aromatic, concentrated and complex mixtures of volatile nonpolar compounds extracted from plant material [19,20]. These natural products have a long history of medical application in humans and are often utilized in many medical practices around the world. The medicinal properties of EOs include anti-inflammatory, antiseptic, anticancer, antispasmodic (and other positive impacts on the gastrointestinal tract), and even anti-anxiety properties, along with antibacterial, antifungal and antiparasitic effects [17,20,21,22,23]. In veterinary medicine, EOs are increasingly used for the treatment and prevention of various diseases, but are still preferable in monogastric animals such as pigs and poultry [22,24,25,26,27]. Given their shown in vivo efficacy, they are also reflected in the many uses of EOs and their formulations in dogs and cats, i.e., against skin diseases caused by *Malassezia pachydermatis* [28], *Otodectes cynotis* [29], various ticks (*Ixodes* spp. and *Rhipicephalus sangiuneus*) [30] or endoparasitic diseases such as *Ancylostoma canimum* [31]. However, some implications and reports suggest their possible use as anthelmintic agents also in ruminants, mainly against GINs [13,18,19,32]. EOs obtained from plants such as *Thymus vulgaris*, *Rosmarinus officinalis*, *Lavandula officinalis*, *Eucalyptus* spp., *Cymbopogon* spp. *Melaleuca alternifolia*, *Lippia sidoides*, *Croton zehntneri*, *Citrus sinensis* etc, and their isolated compounds such as carvacrol, thymol, anethol, eucalyptol, eugenol, citral and others, are among the most referenced [11,14,15,18,32]. However, an insufficient number of such studies, as well as the fact that there are still many unknowns related to the anthelmintic activity of EOs, suggests the need for new studies. 

The task of selecting EOs for anthelmintic studies is not simple given the large number of plant species from which they can be extracted. However, it is the first step in the revealing of new anthelmintic means and, therefore, should be based on relevant operational strategies and ethnopharmacological/chemotaxonomic data [33]. Therefore, the simultaneous testing of different EOs may contribute to finding and selecting the most appropriate ones for wider evaluation and additional studies. Keeping this in mind, this study aimed to evaluate the in vitro ovicidal effect of 11 different EOs (*Thymus vulgaris* L.—two types, *Achillea millefolium* L.—two types, *Satureja montana* L., *Satureja hortensis* L., *Mentha x piperita* L., *Foeniculum vulgare* L., *Helichysum arenarium* L., *Origanum vulgare* L., and *Juniperus communis* L.) and one binary combination of isolated EO compounds (linalool:estragole) against sheep GINs using the egg hatch test (EHT). The aim of this study was also to evaluate the in vivo anthelmintic potential of *T. vulgaris* EO type 1 and linalool:estragole binary combination using the faecal egg count reduction test (FECRT) against GINs in sheep farms in order to estimate the possibility of their application in field conditions.

## 2. Materials and Methods

### 2.1. Essential Oils and Chemical Analyses

In the present study, 11 different essential oils and one binary combination of isolated compounds were used. Samples were purchased from the following producers: *Satureja montana* and *Achillea millefolium* L. (type 1) from Institute of Field and Vegetable Crops, Novi Sad, Serbia; *Foeniculum vulgare* Mill., *Achillea milefolium* L. (type 2), *Mentha × piperita* L. and *Helichrysum arenarium* (L.) Moench from BIOSS, Serbia; *Origanum vulgare* L. from Hippocratic Essentials P.C., Neo Perivoli, Greece; *Thymus vulgaris* L. (type 1) and binary combination of linalool:estragole from Alekpharm, Belgrade, Serbia; *Thymus vulgaris* L. (type 2) from Albert Vieille Sas, Vallauris, France; and *Satureja hortensis* L. From Bio Salas Farago, Orom, Serbia.

Qualitative and semi-quantitative chemical characterization of the essential oils was performed by gas chromatography-mass spectrometry (GC-MS) analyses, whereby the Agilent Technologies 6890 N gas chromatograph coupled with the Agilent Technologies 5975B electron ionization mass-selective detector were used. The analyses were done under the following technical conditions: injection volume of EO 1 μL; injector temperature 250 °C; split ratio 1:10; carrier gas helium; flow rate: 1 mL/min; capillary column: HP-5 (30 m × 0.25 mm, 0.25 μm); temperature program 50–270 °C; ion source temperature 230 °C; electron energy 70 eV; and quadrupole temperature 150 °C. Data were acquired in scan mode (*m*/*z* range 35–400) with a solvent delay of 2.30 min and processed using Agilent Technologies MSD ChemStation software (revision E01.01.335) combined with AMDIS (ver. 2.64) and NIST MS Search (ver. 2.0d) (Agilent Technologies, Inc., Santa Clara, CA, USA). The identification of compounds was done by comparison of mass spectra with data libraries (Wiley Registry of Mass Spectral Data, 7th ed. and NIST/EPA/NIH Mass Spectral Library 05) and confirmated by comparison of linear retention indices with literature data [34]. The relative amounts of components, expressed in percentages, were calculated by the normalization procedure according to the peak area in the total ion chromatogram.

### 2.2. Egg Hatch Test

The in vitro EHT was performed at the Regional Centre for Monitoring of Parasitosis (CREMOPAR), located in Eboli (Salerno), Italy in 2019. GIN eggs were recovered from faecal samples collected from the rectal ampulla of sheep with a natural-mixed infection on two farms located in that region. The samples were processed within 2 h of collection by using the recovery technique as described by Bosco et al. (2018) [35]. Firstly, faecal samples were homogenized and filtered under running water through meshes of different mesh sizes (1 mm, 250, 212 and 38 μm) to separate the nematode eggs from the faeces. Eggs retained on the last sieve were then washed and centrifuged for 3 min at 1500 RPM with distilled water, after which the supernatant was discarded. Following that, centrifugation was performed using a 40% sugar solution to float the eggs, which were then isolated into new tubes and mixed with distilled water. In the end, obtained solutions were centrifuged two more times to remove pellets and to obtain an aqueous solution with GIN eggs.

The EHT procedure was performed as described by Ferreira et al. (2018) [33], with some exceptions. Twenty-four-well plates containing aqueous solutions of approximately 150 eggs/well were used for this experiment as follows: six concentrations of each tested EO sample (50, 12.5, 3.125, 0.781, 0.195 and 0.049 mg/mL) were emulsified in Tween 80 (3%, *v/v*) and completed with distilled water in a final volume of 0.5 mL/well. The preparations were put on the incubator for 48 h at 27 °C, after which the number of eggs and first-stage larvas (L_1_) were counted under an inverted microscope. The obtained values were expressed as the mean percentage of the egg hatching inhibition. The positive control was thiabendazole at a concentration of 0.025 mg/mL, whilist the negative control was 3% Tween 80, *v/v*. The experiment was conducted twice with two replicates each, whereby obtained values were expressed as an arithmetic mean.

### 2.3. Faecal Egg Count Reduction Test

The in vivo FECRT was performed on the same two farms that were used for the EHT (Salerno). 

The animals used for this study were mainly the Lacaune/Bagnolese mixedbreed dairy sheep, homogeneous in age (2 years ± 0.5), body weight (55 kg ± 0.5) and grazing season, without any prior anthelmintic treatments in at least 6 months.

After the confirmation of natural-mixed infection by GINs using the faecal egg count, on both examined farms a total of 48 sheep were divided into four groups (*n* = 12 per group) as follows: 

Group 1 (G1): treated P.O. (oral administration) with 100 mg/kg of body weight of *T. vulgaris* type 1 EO as single dose; 

Group 2 (G2): treated P.O. with 100 mg/kg of body weight of linalool:estragole; 

Group 3 (G3): treated P.O. with 5.0 mg/kg of body weight fenbendazole (positive control); 

Group 4 (G4): treated P.O. with 50 mL sunflower oil (negative control). 

Both *T. vulgaris* type 1 EO and linalool:estragole were diluted in sunflower oil before administration to avoid their effects on mucosa of the gastrointestinal tract. Individual faecal samples were collected rectally on the day of treatment (D0) and after 7 and 14 days (D7 and D14), and stored shortly thereafter at 4 °C. The faecal samples were analysed individually using the Mini-FLOTAC technique with a detection limit of 5 eggs per gram (EPG) of faeces, using a sodium chloride flotation solution (specific gravity = 1.200), as described by Cringoli et al. (2017) [36]. The results were expressed as EPG in each sample with the calculation of averages within each group, and the reduction in GIN eggs in faeces after treatment was evaluated on D7 and D14 after treatment. Final results are presented as arithmetic means from two farms. 

### 2.4. Coproculture

The equal quantity of faeces was collected from each sample before storage at the temperature of 4 °C, in order to create a pool for each faecal culture group at different time points (D0, D7 and D14). The procedure was done according to the protocol described by the Ministry of Agriculture, Fisheries and Food of UK in 1986 [37]. Developed third-stage larvae (L_3_) were identified based on their morphological specifics, as suggested by van Wyk and Mayhew [38]. Identification and percentages of each nematode genera were conducted on 100 L_3_, whereby all larvae were identified if a sample had 100 or less L_3_ present. In this way, of the total number of larvae identified, it was possible to obtain the percentage of each genus. 

### 2.5. Statistical Analyses

The percentages of egg hatch inhibition were calculated using a formula proposed by Coles et al. (1992) [39]:EHT = [(number of eggs)/(number of larvae + number of eggs)] × 100

For the comparation of values obtained for different concentrations and the controls within one EO, one-way Analysis of Variance (ANOVA) followed by Tukey’s test (*p* < 0.05) was performed. On the other hand, for the comparation of the values of the same concentration between different EOs, two-way ANOVA followed by Tukey’s test (*p* < 0.05) was used. Nonlinear regression/logarithmic distributions were applied for the calculation of half-maximal inhibitory concentrations (IC_50_) [33].

The reduction in faecal egg counts (FECR) in animal’s faeces was calculated by using a formula described by Macedo et al. (2010) [40]:FECR (%) = 100 × (1 − T_2_/T_1_ × C_1_/C_2_) 
where T_1_ is EPG before treatment in treatment groups, T_2_ is EPG after treatment (day 7 or 14) in treatment groups; C_1_ is EPG before treatment in the negative control group; C_2_ is EPG after treatment in the negative control group.

The obtained values were analyzed and compared by using Two-way ANOVA followed by Tukey’s test (*p* < 0.05).

Two-way ANOVA (*p* < 0.05) was performed also in the anaylsis of the results of coproculture to evaluate the differences in the ratio between the percentage of GIN genera found pre- and post-treatment.

Statistical analyses were performed using the program GraphPad Prism 9.2.0. (GraphPad Holdings, LLC, San Antonio, CA, USA)

## 3. Results

### 3.1. GC-MS Analyses

The results of the conducted GC-MS analyses reveal the rich chemical composition of tested EO samples, containing a wide number of ingredients with possible anthelmintic properties (Table 1). The ratio of linalool:estragole in the tested binary combination was 19%:81%.

### 3.2. Egg Hatch Test

The obtained EHT results showed the great anthelmintic potential of tested EO samples (Table 2). Three EOs, *O. vulgare*, *S. montana* and *F. vulgare* were 100% effective in the inhibition of egg hatchability at each tested concentration. Two samples of *T. vulgaris* and *S. hortensis* EOs were also highly effective with an egg hatch inhibition of 95.3–100%, 98.5–100% and 99.3–100%, respectively, showing the similar anthelmintic activity to that of thiabendazole, 98.0% (*p* > 0.05). The rest of the EOs showed medium, dose-dependent activity: *J. communis* 81–96.8% (R^2^ = 0.9442), *M. piperita* 72.5–99.8% (R^2^ = 0.9834), two types of *A. millefolium* 46.5–100% (R^2^ = 0.9840) and 69.5–97.3% (R^2^ = 0.8637), respectively and linalool:estragole 29.5–100% (R^2^ = 0.9528). The least potential was shown by *H. arenarium* EO with an efficacy of 59.8–69.3%. The calculated IC_50_ values for *T. vulgaris* type 2, *S. hortensis*, *M. piperita*, *J. communis*, *A. millefolium* type 1, *H. arenarium* and linalool:estragole were 0.098, 0.187, 0.281, 0.495, 0.517, 0.952 and 0.980 mg/mL, respectively. 

### 3.3. Faecal Egg Count Reduction Test

Two tested samples, the *T. vulgaris* EO and the binary combination of linalool:estragole showed similar in vivo anthelmintic potential with a reduction in counted nematode eggs in faeces of 25.23% and 24.91% on Day 7 and 24.42% and 25.90% on Day 14, respectively (at flock level). However, EPG values do not statistically differ from sunflower oil (*p* > 0.05), although on Day 7 the EPG values of tested samples were similar to that of fenbendazole (*p* > 0.05). On Day 14, the counted EPG values of fenbendazole were significantly lower (*p* < 0.05) than that of tested samples (Table 3). Based on individual FECRT, the average percentage of individual EPG reduction in a groups of *T. vulgaris* and linalool:estragole reached 39.30% and 51.88% on Day 7, respectively, although it was less than 10% on Day 14. Individual FECRTs for fenbendazole were 84.73% and 87.32% on Days 7 and 14, respectively. Animals were observed for the presence of side effects after application of EOs, without any toxic effects observed.

### 3.4. Coproculture

Four genera of sheep GINs were identified on coproculture examination on both tested farms. In total, their representation on Day 0 was as follows: *Haemonchus* 53%, *Trichostrongylus* 29.5%, *Teladorsagia* 14.5% and *Chabertia* 3%. After treatment, their percentages changed somewhat differently depending on the treatment group, whereby in most groups the % of *Haemonchus* decreased and *Trichostrongylus* increased on days 7 and 14. The % of *Teladorsagia* increased in the control groups, while the % of *Chabertia* increased in the group treated by *T. vulgaris* and decreased in the group treated by linalool:estragole. However, neither these nor changes observed in the aspect before and after treatment were statistically significant (*p* > 0,05) in any of the tested groups. The percentages of each GIN genera in each in vivo treatment group after treatment (Day 7 and 14) are shown in Figure 1A–D. 

## 4. Discussion

The use of plant products such as essential oils in anthelmintic treatments has many advantages. Firstly, EOs have a rich chemical composition of various bioactive compounds with huge pharmacological potential [14], which can lead to high activity against nematodes, as shown in our in vitro study. Furthermore, a large number of compounds in EOs that belong to different chemical classes may contribute to a reduced susceptibility to resistance [14,41]. Some reports state that botanical anthelmintics may be considered well tolerated by animals from the toxicology perspective and are related to the low amount of residues in meat and milk [33]. Although this statement is not proven and requires exact studies upon this, a large number of EOs, as well as their ingredients belonging to different chemical groups, offers an opportunity to find those who best meet these requirements, along with appropriate efficacy. This also refers to the price, which vary depending on the EO. Finally, their easy accessibility in regions with developed biodiversity allows easy acquisition [33] and, along with other mentioned factors, offers the possibility of a sustainable avenue for nematode control in ruminants.

Evaluation of the anthelmintic efficacy of EOs requires the application of appropriate methods. In vitro studies, wherein the EHT is one of the most frequently used and recommended tests [39], have an advantage due to their speed, low cost, high reproducibility, ease of application and lack of experimental animals [33]. Therefore, these tests are very useful for the initial evaluation of anthelmintic potential in new active substances, which in turn leads to further studies [41]. However, in vivo studies, whereby an FECRT is the method of choice for monitoring of anthelmintic efficacy [42], provide clearer data on efficacy and the possibilities for application in practice. Thus, the results of laboratory and field conditions testing do not necessarily coincide, whereby obtained efficacy is usually higher in laboratory testing [14]. This finding may be explained by many factors that can affect in vivo activity, which is especially true for EOs due to their unstable nature [20]. Therefore, both in vitro and in vivo tests are important in the process of developing new anthelmintic drugs [43].

In the present study, practically all tested samples demonstrated high ovicidal anthelmintic potential. The highest inhibitory effect on egg hatchability was shown by EOs of *O. vulgare*, *F. vulgare*, *S. montana*, *S. hortensis* and both *T. vulgaris* samples wherein all tested concentrations showed either a maximum effect or activity higher than 95%. The criteria for in vitro testing, whereby samples with efficacy greater than 90% can be considered effective for the control of nematodes including GINs [14,40,41], was set up by the World Association for the Advancement of Veterinary Parasitology (WAAVP). According to these criteria, mentioned EOs may present a very promising research subject for further studies. This also refers to EOs *J. communis* and *M. piperita*, since most of their concentrations were more than 90% efficient as well, although the activity of these EOs was dose-dependent. Two types of *A. millefolium* EOs and the binary combination of linalool:estragole also expressed dose-dependent activity with a higher efficacy at high concentrations (*p* > 0.05 compared to the thianbendazole) and medium or low activity at lower concentrations, which makes them suitable for further examinations as well.

The high obtained values of standard deviation for mean EPGs in FECRTs suggest large differences in worm burdens in animals. This can be explained by the fact that in small ruminants, gastrointestinal parasites are highly aggregated and over-dispersed within the host, whereby most of the parasite population (approximately 80%) is found in only 20–30% of hosts, while the majority of animals have low worm burdens [4,44]. In any case, two tested EO formulations showed a degree of anthelmintic potential with a reduction in the number of counted eggs in faecal samples of approximately 25% each day of testing (7 and 14 days after treatment). Although the positive control was clearly more effective, neither of them reach the needed efficacy of 90% recommended by the WAAVP for in vivo testing [42], which indicates developed resistance to benzimidazoles to some extent. The limited efficacy of the *T. vulgaris* EO type 1 and linalool:estragole may be explained by the fact that EOs and their active ingredients are prone to evaporation and destabilization [20,45], which makes them degradable in animal organisms as mentioned above. This particularly refers to ruminants keeping in mind the anatomical and physiological specifics of their gastrointestinal tract, which may affect perorally applied active substances [46]. Thus, active substances of tested EO formulations were most likely partially deactivated prior to reaching the targeted place of action in the abomasum and small intestine. 

However, these obstacles may be circumvented by different encapsulation techniques that may protect the active substances from degradation and allow their better bioavailability and higher in vivo efficacy. Also, encapsulation may enable controlled release of active ingredients of EOs, as well as reduce their smell and taste that could be unpleasant for animals [20,24,44,47]. Another method is to change the means of application, such as to use lick blocks containing plant-based compounds that may provide long-term use. In some research, for example, these preparations effectively reduced coccidian invasion and had a beneficial effect on growth and body development of the lambs [48]. Nevertheless, the finding that there were no observed toxic effects on tested animals allows an opportunity for an increased dose in future trials to reach a higher efficacy. The obtained results and demonstrated efficacy should not be neglected since they suggest that these plant products, coupled with other methods, may still play a significant role in an integrated approach for nematode control in ruminants, in case they are proven ineffective as independent products [40,41]. Furthermore, our research group has already made new experimental efforts with the evaluation of the in vivo efficacy of other highly effective in vitro EOs from the present study (data not shown). 

The results of coprocultre showed the presence of four GIN genera on the examined farms: *Haemonchus*, *Trichostrongylus*, *Teladorsagia* and *Chabertia* in different percentages. Although some changes in their percentages after treatment were observed while comparing different treatment groups, the results obtained from the cultures of all groups did not show any significant difference in the ratio between the percentage of genera found prior and post treatment. This result suggests that none of the employed treatments are specific for only a single genera. 

The *T. vulgaris* EO was previously in vitro and in vivo tested for anthelmintic activity against sheep GINs. In a study by Ferreria et al. (2016), it also exhibited high ovicidal activity, where the concentrations ranging from 0.097–50 mg/mL inhibited the hatchability of *H. contortus* eggs by 49.4–100% with an IC_50_ of 0.436 mg/mL [14]. However, in the same study, the tested EO failed to reduce the EPG at doses of 75, 150 and 300 mg/kg. The differences between these results and the results obtained in the present study may be explained by the differences in the chemical composition of used samples of *T. vulgaris*, as will be discussed later. On the other hand, the *M. piperita* EO was previously investigated in a study of Katiki et al. (2011), where the calculated IC_50_ value was 0.26 mg/mL [49], similarly as in the present study. The rest of the samples tested in the present study are new, according to the best of our knowledge. 

As discussed earlier, the anthelmintic activity of EOs originates from their rich chemical composition and many bioactive ingredients belonging to different chemical classes [50]. However, ingredients of EOs can significantly differ in their degree of anthelmintic activity [15], suggesting the importance of compounds present in EOs and their percentages. The GC-MS analyses performed in the present study showed that the presence of compounds such as carvacrol, anethole, thymol, p-cymen and y-terpinene was related to the heightened efficacy of tested samples, since the most effective EOs were composed of these ingredients. Indeed, the high activity of isolated anethole, carvacrol and thymol against *H. contortus* eggs was demonstrated in a study by Katiki et al. (2017) in EHT with obtained IC_50_ values of 0.07, 0.11 and 0.13 mg/mL, respectively [15]. The ovicidal activity of thymol was also demonstrated in other studies, with obtained IC_50_ values of 0.442 [14] and 0.08 mg/mL [51], as well as carvacrol and anethol with IC_50_ values of 0.17 [52] and 0.69 mg/mL [53], respectively. According to the best of our knowledge, the activity of isolated p-cymen and y-terpinene against sheep GINs is still unconfirmed, although these ingredients were found in EOs that demonstrated high anthelmintic activity against these parasites, such as γ-terpinene in *Melaleuca alternifolia* (20.15%) [54].

The results of the conducted chemical analyses also suggest that the level of efficacy of EOs does not correspond with the number of identified compounds in tested samples. In contrast, the most efficient EOs such as *O. vuglare*, *F. vulgare, S. montana* and *S. hortensis* had the lowest number of identified compounds, while the least effective EOs, *H. arenarium* and two samples of *A. millefolium*, had the highest number of components. This indicates that not all components of EOs possess anthelmintic potential, but usually the main or two or three dominant compounds [55]. However, the presence of other compounds could contribute to the overall activity by synergistic action through a different mechanism; thus whole EOs usually express better efficacy than their isolated compounds when applied individually [14,53].

Finally, many factors can affect the chemical composition of EOs and thus their biological activities. These include various exogenous factors such as light, precipitation, growth site, soil properties (hydrology, pH and salinity), seasonal variation and endogenous factors such as the location of production and accumulation of the EOs in the plant, the age of the plant, as well as the genetic characteristics that regulate their secondary metabolism [56,57]. Additionally, biotic factors such as the presence of certain soil organisms and microorganisms may also be included [57]. All these factors lead to differences in the composition of EOs obtained from different plant species, but also in the same plant species as shown in the present study. Namely, the differences in chemical composition within two samples of *T. vulgaris*, as well as *A. millefolium*, EOs have led to the differences in their ovicidal activity, which were significant (*p* < 0.05) for some concentrations. Those differences are most likely derived from different plant varieties used for EO isolation by producers and, on the other hand, the different geographical origin of plants (in the case of *T. vulgaris*, samples were obtained from Serbia and France; *A. millefolium* samples were obtained from two different producers from Serbia).

All the previously mentioned findings related to the chemical composition of EOs should not be neglected and may be highly important to the pharmaceutical industry from the aspect of finding an appropriate EO-based formulation designed to control GINs in ruminants. This also indicates the importance of conducting GC-MS analyses in studies such as this before the selection of samples for testing and conduction of in vitro and in vivo tests.

Understanding the mechanism of action of EOs and their ingredients is important for their practical use in nematode control, given that it can ensure that useful information on the most appropriate formulation and delivery means is obtained [58]. However, current understanding is still inadequate and there are efforts in this field to improve upon this state. So far, numerous mechanisms of nematocidal activity of EOs are proven or suggested depending on the ingredients that make up their composition. These involve interruption of the nematode nervous system, the inhibition of AChE activity, interference with the neuromodulator octopamine or GABA-gated chloride channels, disruption of the cell membrane of the nematode thereby changing its permeability, membrane and ion channel perturbations modifying membrane-bound protein activity and the intracellular signalling pathways, etc. [58]. The activity of phenolic compounds, such as carvacrol and thymol, may be associated with damage caused to the cuticle and digestive apparatus on nematode larvae. Additionaly, these compounds are most likely related to the neurotoxic effect on the free-living nematodes, since they interact with SER-2 tyramine receptors. The cuticular changes and possible neurotoxicity may interfere with the permeability of the cuticle and motility, hindering the maintenance of homeostasis within these parasites [51,52]. Therefore, different ingredients of EOs induce different neurological and structural changes in nematodes, which may give rise to their paralysis and death.

The idea of the use of EOs for various purposes, including anthelmintic, is relatively new and most data relating to their use in animals are still based on anecdotal observations without scientific validation [59], which are not enough for their wider use in nematode control. However, although there is still a lack of data on the anthelmintic efficacy of EOs, there has been a steady increase in studies aiming to verify the anthelmintic potential of plants [59]. The lack of studies is especially noticable in relation to in vivo testing [18], whereby the results of trials conducted so far, including the present study, show the inferior activity of EOs compared to synthetic drugs [40]. However, finding an appropriate formulation (plant species, chemical composition, dose and method of application) is not a simple task, thus requiring a larger number of studies and works [60,61,62], whereby the comparison between results obtained for different EO samples is also important for the selection. Finally, novel encapsulation methods offer the possibility of overcoming these problems and achieving appropriate efficiency, as discussed earlier.

## 5. Conclusions

An efficient alternative to commercial anthelmintics is urgently needed due to the development of resistance and the economic losses it entails. Therefore, simultaneous testing of different EO samples may contribute to finding valuable anthelmintic agents. The present study demonstrated the high in vitro, but also, to some extent, in vivo, anthelmintic potential of the tested natural-based formulations against sheep GINs. From this perspective, the highest potential was shown by *O. vulgare*, *F. vulgare*, *S. montana*, two samples of *T. vulgaris* and *S. hortensis*, EOs containing compounds such as carvacrol, thymol, anethol, p-cymene and y-terpinene in high percentages.. However, an additional in vivo trial with modified conditions (higher doses, a different means of application or the use of encapsulation methods) for samples tested in vivo (*T. vulgaris* EO and linalool:estragole combination), or a new trial with other mentioned EOs should be conducted to reach higher efficacy in field conditions. Nevertheless, this study is another confirmation of the possible role of botanical anthelmintics in the control of sheep GINs. Thus, and looking from a future perspective, the results of the present study may be of practical importance in combating anthelmintic resistance.

## Figures and Tables

**Figure 1 vetsci-09-00093-f001:**
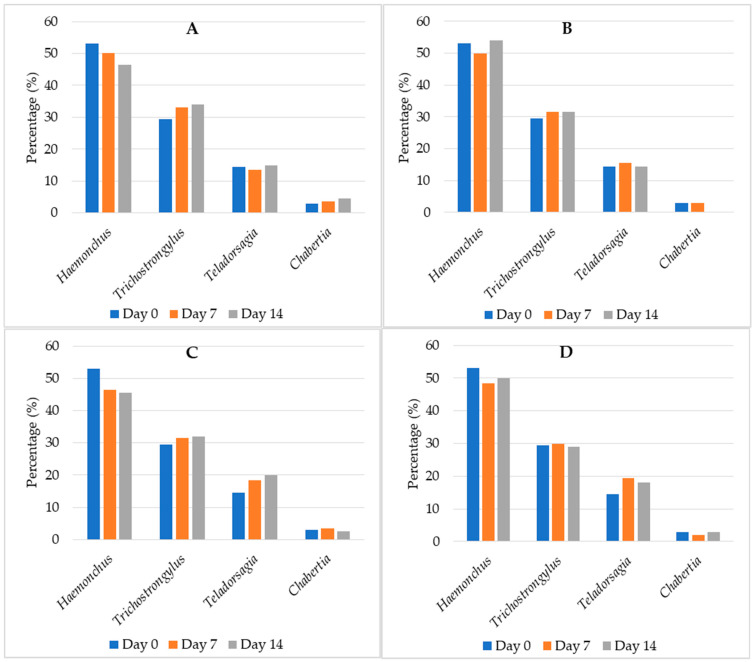
Percentage (%) of sheep GIN genera on tested sheep farms pre and post treatment in each treatment group: (**A**)—G1, *T. vulgaris* EO; (**B**)—G2, linalool:estragole; (**C**)—G3, fenbendazole; (**D**)—G4, sunflower oil.

**Table 1 vetsci-09-00093-t001:** Chemical composition (% of total peak area) of the investigated essential oils determined by GC-MS.

AI	Compound	% of Total Peak Area ^a^										
		TV1	TV2	AM1	AM2	SM	SH	MP	FV	HA	OV	JC
925	α-Thujene	-	0.85	-	0.13	0.33	0.38	-	-	-	-	**1.88 ^b^**
932	α-Pinene	**2.47**	0.71	**3.09**	**4.15**	**1.16**	**1.21**	0.20	**1.53**	**28.0**	**1.81**	**40.46**
947	Camphene	0.62	0.42	**1.37**	0.26	0.44	0.46	-	0.06	0.17	-	0.33
972	Sabinene	-	-	**1.87**	**5.48**	-	-	**1.28**	-	-	-	**14.04**
976	β-Pinene	0.18	0.10	**1.50**	**28.5**	0.66	-	0.12	0.05	0.21	**1.64**	**2.70**
990	β-Myrcene	0.71	**1.05**	-	-	0.96	**1.30**	0.84	0.13	-	0.32	**8.87**
998	δ-2-Carene	-	-	**1.40**	-	-	-	-	-	-	-	-
1016	α-Terpinene	-	**1.18**	**2.86**	0.19	**2.32**	**2.00**	-	-	0.11	0.20	0.86
1024	*p*-Cymene	**41.7**	**21.0**	**4.38**	0.65	**42.8**	**12.9**	0.07	-	-	**12.6**	**1.94**
1026	*o*-Cymene	-	**1.17**	-	-	-	0.54	-	-	-	-	-
1027	Limonene	**1.26**	-	-	0.87	**1.52**	-	**7.14**	**1.20**	**1.66**	-	**4.95**
1030	1,8-Cineole	0.66	-	**41.69**	**11.7**	0.73	-	**2.64**	-	0.57	-	-
1057	γ-Terpinene	-	**8.11**	**1.13**	0.40	**14.6**	**29.7**	**4.46**	0.29	0.31	**2.63**	**1.42**
1059	Artemisia ketone	-	-	**4.31**	-	-	-	-	-	-	-	-
1082	Artemisia alcohol	-	-	**1.57**	-	-	-	-	-	-	-	-
1088	Fenchone	-	-	-	-	-	-	-	**19.5**	-	-	-
1100	Linalool	**4.37**	**2.77**	-	-	**1.20**	-	-	-	0.30	**1.43**	-
1106	cis-Thujone	-	-	**3.28**	-	-	-	-	-	-	-	-
1115	trans-Thujone	-	-	**2.13**	-	-	-	-	-	-	-	-
1124	Chrysanthenone	-	-	**2.55**	-	-	-	-	-	-	-	-
1143	Camphor	0.22	-	**8.37**	**1.57**	-	-	-	0.15	-	-	-
1152	Menthone	-	-	-	-	-	-	**3.33**	-	-	-	-
1163	Isomenthone	-	-	-	-	-	-	**6.04**	-	-	-	-
1164	Borneol	0.69	**1.24**	**3.57**	0.47	**1.27**	-	-	-	-	-	-
1176	Terpinen-4-ol	-	0.62	**3.37**	-	0.78	-	**7.88**	-	-	-	**2.85**
1190	α-Terpineol	**11.7**	0.26	**1.18**	0.87	-	-	**9.77**	-	0.33	-	-
1198	Estragole	-	-	-	-	-	-		**3.37**	-	-	-
1204	trans-Dihydrocarvone	-	-	-	-	-	-	**14.6**	-	-	-	-
1214	Isodihydrocarveol	-	-	-	-	-	-	**6.25**	-	-	-	-
1234	trans-Chrysanthenyl acetate	-	-	**4.90**	-	-	-	-	-	-	-	-
1253	Piperitone	-	-	-	-	-	-	**25.4**	-	-	-	-
1286	Anethol	-	-	-	-	-	-	-	**73.4**	-	-	-
1291	Lavandulyl acetate	-	-	-	**1.30**	-	-	-	-	-	-	-
1292	Thymol	**31.6**	**54.5**	-	-	-	0.36	-	-	-	0.97	-
1302	Carvacrol	-	**3.95**	-	-	**28.1**	**49.5**	-	-	-	**76.2**	-
1364	Neryl acetate	-	-	-	-	-	-	-	-	**3.05**	-	-
1375	α-Copaene	-	-	-	-	-	-	-	-	**3.09**	-	0.28
1384	β-Bourbonene	-	-	-	**1.29**	-	-	0.51	-	-	-	-
1391	β-Elemene	-	-	-	-	-	-	-	-	-	-	**1.24**
1402	*iso*-Italicene	-	-	-	-	-	-	-	-	**3.20**	-	-
1418	β-caryophyllene	0.82	**1.69**	0.59	**18.7**	**2.46**	**1.25**	**1.69**		**6.36**	**2.23**	**1.71**
1442	*sesquiterpene*	-	-	-	-	-	-	-	-	**2.09**	-	-
1452	α-Humulene	-	-	-	**4.08**	-	-	-	-	-	-	**1.31**
1474	*sesquiterpene*	-	-	-	-	-	-	-	-	**1.64**	-	-
1479	γ-Curcumene	-	-	-	-	-	-	-	-	**20.1**	-	-
1480	Germacrene D	-	-	0.44	**8.01**	-	-	-	-	-	-	**2.54**
1482	ar-Curcumene	-	-	-	-	-	-	-	-	**4.15**	-	-
1485	β-Selinene	-	-	-	-	-	-	-	-	**9.32**	-	-
1493	α-Selinene	-	-	-	-	-	-	-	-	**5.21**	-	-
1499	*sesquiterpene*	-	-	-	-	-	-	-	-	**1.33**	-	-
1506	*sesquiterpene*	-	-	-	-	-	-	-	-	**1.33**	-	-
1512	β-Curcumene	-	-	-	-	-	-	-	-	**1.89**	-	-
1513	γ-Cadinene	-	-	-	-	-	-	-	-	-	-	**1.09**
1523	δ-Cadinene	-	-	-	0.90	-	-	-	-	**1.53**	-	**2.46**
1556	Germacrene B	-	-	-	-	-	-	-	-	-	-	**2.32**
1582	Caryophyllene oxide	-	-	-	**2.49**	-	-	**2.14**	-	-	-	-
1590	Viridiflorol	-	-	-	**3.52**	-	-	-	-	-	-	-
Number of all identified compounds		15	19	28	27	17	13	21	12	30	10	28

AI—arithmetic retention index;—not detected; TV1—*Thymus vulgaris* (Serbia); TV2—*Thymus vulgaris* (France); AM1—*Achillea milefolium* type 1; AM2—*Achillea milefolium* type 2; SM—*Satureja montana*; SH—*Satureja hortensis*; MP—*Mentha x piperita*; FV—*Foeniculum vulgare*; HA—*Helichrysum arenarium*; OV—*Origanum vulgare*; JC—*Juniperus communis*; ^a^ Only the compounds present in more than 1% in at least one essential oil are presented in the table; ^b^ Compounds with abundance >1% are written in bold.

**Table 2 vetsci-09-00093-t002:** Inhibition of egg hatchability (mean ± standard deviation) of sheep gastrointestinal nematodes at different concentrations of tested essential oils.

Concentration [mg/mL]	*Thymus vulgaris* 1	*Thymus vulgaris* 2	*Achillea millefolium* 1	*Achillea millefolium* 2	*Satureja montana*	*Satureja hortensis*
50	100 ± 0 ^Aa^	100 ± 0 ^Aa^	99.5 ± 1.0 ^Aa^	97.3 ± 0.96 ^Aa^	100 ± 0 ^Aa^	100 ± 0 ^Aa^
12.5	100 ± 0 ^Aa^	99.5 ± 0.58 ^Aa^	98.0 ± 1.83 ^Aa^	90.0 ± 4.97 ^Ab^	100 ± 0 ^Aa^	100 ± 0 ^Aa^
3.125	95.3 ± 5.68 ^Aa^	100 ± 0 ^Aa^	95.3 ± 4.35 ^Aa^	73.0 ± 1.63 ^Bb^	100 ± 0 ^Aa^	100 ± 0 ^Aa^
0.781	97.5 ± 1.73 ^Aae^	100 ± 0 ^Aa^	87.5 ± 2.65 ^Bbe^	72.8 ± 8.42 ^Bc^	100 ± 0 ^Aa^	100 ± 0 ^Aa^
0.195	98.0 ± 1.83 ^Aa^	100 ± 0 ^Aa^	49.0 ± 1.63 ^Cb^	71.3 ± 3.3 ^Bc^	100 ± 0 ^Aa^	99.8 ± 0.5 ^Aa^
0.049	96.8 ± 2.22 ^Aa^	98.5 ± 0.58 ^Aa^	46.5 ± 3.0 ^Cb^	69.5 ± 3.7 ^Bc^	100 ± 0 ^Aa^	99.3 ± 0.96 ^Aa^
Control (+)	98.0 ± 0.82 ^A^	98.0 ± 0.82 ^A^	98.0 ± 0.82 ^A^	98.0 ± 0.82 ^A^	98.0 ± 0.82 ^A^	98.0 ± 0.82 ^A^
Control (−)	16.8 ± 5.56 ^B^	16.8 ± 5.56 ^B^	16.8 ± 5.56 ^D^	16.8 ± 5.56 ^C^	16.8 ± 5.56 ^B^	16.8 ± 5.56 ^B^
	** *Mentha x piperita* **	** *Foeniculum vulgare* **	** *Helichrysum arenarium* **	** *Origanum vulgare* **	** *Juniperus communis* **	** *Linalool: estragole* **
50	99.8 ± 0.5 ^Aa^	100 ± 0 ^Aa^	69.3 ± 2.22 ^Ab^	100 ± 0 ^Aa^	96.8 ± 1.71 ^Aa^	100 ± 0 ^Aa^
12.5	99.0 ± 0.82 ^Aa^	100 ± 0 ^Aa^	68.5 ± 2.89 ^Ac^	100 ± 0 ^Aa^	95.5 ± 1.73 ^ABab^	100 ± 0 ^Aa^
3.125	99.0 ± 0.82 ^Aa^	100 ± 0 ^Aa^	68.3 ± 3.59 ^Ab^	100 ± 0 ^Aa^	94.8 ± 0.96 ^ABa^	99.8 ± 0.5 ^Aa^
0.781	94.8 ± 1.71 ^Aae^	100 ± 0 ^Aa^	63.8 ± 1.26 ^ABd^	100 ± 0 ^Aa^	91.0 ± 1.63 ^Be^	47.0 ± 20. 5 ^Bf^
0.195	83.0 ± 1.63 ^Bd^	100 ± 0 ^Aa^	59.8 ± 2.22 ^Be^	100 ± 0 ^Aa^	85.5 ± 0.58 ^Cd^	29.5 ± 1.29 ^BCf^
0.049	72.5 ± 1.29 ^Cc^	100 ± 0 ^Aa^	59.8 ± 0.96 ^Bd^	100 ± 0 ^Aa^	81.0 ± 1.63 ^Ce^	29.5 ± 2.65 ^BCf^
Control (+)	98.0 ± 0.82 ^A^	98.0 ± 0.82 ^A^	98.0 ± 0.82 ^A^	98.0 ± 0.82 ^A^	98.0 ± 0.82 ^A^	98.0 ± 0.82 ^A^
Control (−)	16.8 ± 5.56 ^D^	16.8 ± 5.56 ^B^	16.8 ± 5.56 ^C^	16.8 ± 5.56 ^B^	16.8 ± 5.56 ^D^	16.8 ± 5.56 ^C^

Uppercase compares means between different concentrations in one EO and controls; lowercase compares means between same concentrations of different EOs. Different letters indicate significant differences (*p* < 0.05). Control (+)—Thiabendazole, 0.025 mg/mL; Control (−)—3% Tween 80, *v/v*.

**Table 3 vetsci-09-00093-t003:** EPG (mean ± standard deviation) and efficacy of *T. vulgaris* EO and linalool:estragole binary combination based on faecal egg count reduction test (at flock level).

Tested Sample		Day 0	Day 7	Day 14
*T. vulgaris* type 1, 100 mg/kg	Mean EPG	143.5 ± 170.6 ^Aa^	99.8 ± 123.1 ^ABb^	107.7 ± 115.1 ^Aab^
	Efficacy	/	25.23%	24.42%
Linalool:estragole, 100 mg/kg	Mean EPG	145.6 ± 198.2 ^Aa^	101.9 ± 148.2 ^ABb^	106.3 ± 124.8 ^Ab^
	Efficacy	/	24.91%	25.90%
Fenbendazole, 5 mg/kg (C+)	Mean EPG	221.4 ± 326.9 ^Aa^	37.0 ± 86.5 ^Ab^	24.8 ± 39.6 ^Bb^
	Efficacy	/	82.74%	88.93%
Sunflower oil, 50 mL/animal (C−)	Mean EPG	142.3 ± 149.1 ^Aa^	132.5 ± 119.4 ^Ba^	140 ± 100.1 ^Aa^
	Efficacy	/	/	/

Uppercase compares means between different groups at one time point; lowercase compares means of different time points within one group. Different letters indicate significant differences (*p* < 0.05).

## Data Availability

All data generated or analysed during this study are included in this published article. The datasets used and/or analysed during the present study available from the corresponding author upon reasonable request.

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
