# Peer review of "Anthelmintic Properties of Essential Oils to Control Gastrointestinal Nematodes in Sheep—In Vitro and In Vivo Studies"

_vetsci, 2022, doi:10.3390/vetsci9020093_

Round 1

Reviewer 1 Report

Anthelmintic properties of essential oils to control gastrointestinal
nematodes in sheep - in vitro and in vivo studies

The work is interesting and done to good standards. It touches upon an important aspect of searching for new substances with anti-parasite activity.

The author proved the knowledge of the issue and research procedures.

The assessed work is a valuable achievement of its authors.

After analyzing the text, several questions arise.

Are there any other parasites apart from the mentioned parasites in the examined sheep?

If so, how only GIN eggs were isolated.

Were the research groups of animals qualified for the FECRT test randomized?

If so, by what date was the group randomization performed?

 In my opinion, the initial examination should be performed on the date D-7, Therefore, in addition to the dates D0, D7, and D14, the description of the experiment should also include the date D-7

As part of the discussion, I suggest comparing the single use of the preparation with its long-term use in the form of licks - In my opinion, this solution is more effective. Publications to compare this way of using essential oils.

DOI:10.1016/j.smallrumres.2014.11.019

DOI:10.1515/helmin-2016-0008

DOI:10.1016/j.vetpar.2008.10.021

DOI: 10.21521/mw.5806

Reviewer 2 Report

General comments: Given the issues with AR, identifying alternatives and integrated approaches is critical and this work contributes to the knowledge that is needed to develop effective new GIN control approaches. A primary concern is the potential over-selling of the efficacy of EOs within the manuscript with references to high efficacy without consideration of the potency compared to registered anthelmintics. Also, when looking at the results, the control in vivo and the EOs are not significantly different. Hence referring to the EOs as efficacious is questionable. Also, fenbendazole typically is evaluated at 10-14 days post treatment with 7 days potentially underestimating efficacy. Hence comparing day 7 and stating that the EOs are not significantly different is not a fair assessment. A higher dose of EOs could very well achieve efficacy different from sunflower oil and closer to fenbendazole, but that study has yet to be done.

Abstract

  1. Line 20: While some EOs show efficacy, given the level of efficacy compared to expectations with registered anthelmintics, I would hesitate to say they are a promising alternative. Maybe state “present a potential alternative” or “a promising role in GIN control” as suggested in line 346.
  2. Line 26: Instead of “applied perorally” consider “orally administered”
  3. The abstract should state the taxonomic group of the eggs used in the EHT and the in vivo study

Introduction

  1. Lines 56-58: It is unclear what is meant in regards to even higher economic losses. With failed efficacy, losses are even higher than with no treatment for the GIN?
  2. Line 61: consider changing “or” to “and”. With “or” it infers these are alternatives that can be used individually to replace anthelmintics. However, given how these alternatives work and their efficacy, they are used in combinations as indicated on line 346.
  3. Line 63: anthelmintics have; also, it might be needed to define the difference between a botanical and an herbal product
  4. Line 68: history in animals or people or both?
  5. Line 87: it is unclear what is meant by “further steps”

Methods

  1. Line 135: which were then isolated
  2. Out of curiosity, why thiabendazole vs fenbendazole or albendazole as a positive control in the EHT?
  3. When were body weights taken to determine dose?
  4. Line 181: check consistency of L3 vs L3

Results

  1. Line 226: 98.0
  2. Line 253: without any toxic effects observed or with no toxic effects observed
  3. See general comments

Discussion

  1. The discussion is thorough but there are some areas where it might be possible to be more concise. It is a bit of a balancing act, being thorough and providing information on EOs vs being concise.
  2. In the discussion it might be worth mentioning toxicity and residue issues. Give the variety of compounds in an essential oil, identifying which to monitor for residues could be challenging. Also, while consistent composition is mentioned, contaminants (heavy metals for example) are not mentioned as a potential issue.
  3. Also, there are many statements about efficacy being high. However, if you look at the dose compared to the dose of registered anthelmintics, the lowest dose in the EHT is close to the maximum dose used in studies with benzimidazoles. Therefore, the efficacy is actually low in a mg to mg comparison. Hence, the potency compared to benzimidazoles and macrocyclic lactones is low. I think it is important to mention this. I don’t think it is necessarily something that will prevent the use of EOs, but it is quite different to administer 1 ml or 2 ml vs 10 or 20 ml.

References

  1. The formatting of these are inconsistent
  2. Scientific names are not in italics

Reviewer 3 Report

Dear Authors, I found your manuscript very interesting. I'm providing my revision, with some comments and seggenstions along the pdf file. Please, take into consideration some comments along the discussion that I think could give strenght to your manuscritpt. 

Author Response

Dear Reviewer 3, we are very glad that you found our manuscript very interesting. We are grateful for your marks on the quality of our paper. We did our best to additionally improve it in a new version, bearing in the mind your suggestions and ideas. 

Round 2

Reviewer 2 Report

Thank you for addressing my previous concerns and comments.

In some of the additions, a quick review for English grammar is needed to correct minor typos.